# MATH FOR AI: ON THE GENERALIZATION OF LEARNING MATHEMATICAL PROBLEM SOLVING

## ABSTRACT

There has been a growing interest in enhancing the mathematical problem-solving (MPS) capabilities of LLMs. While some researchers focus on developing specialized math models to advance AI for math, others study mathematical reasoning with a *math for AI* perspective, positing that integrating mathematical reasoning data could enable LLMs to perform complex reasoning more broadly. This hypothesis draws from neuroscience studies which show that solving mathematical problems aids in the development of general reasoning skills in humans. The concept of "math for AI" has gained particular relevance as the research community increasingly focuses on complex reasoning – Given the scarcity of complex and lengthy chain-of-thought data, MPS emerges as a prime candidate for collecting or synthesizing substantial volumes of intricate thought processes, thus serving as a potential key resource for enhancing general complex reasoning. However, it remains unclear whether skills acquired through learning MPS can extend to other reasoning tasks or merely improve MPS-specific benchmark scores. In this paper, we present a comprehensive empirical analysis to address this question. Specifically, we explore three prevalent methods for improving MPS: (1) continual pretraining on mathematical text; (2) instruction pretraining on large-scale QA pairs synthesized from raw text; and (3) instruction tuning on MPS datasets. Through controlled experiments and evaluations across seven distinct reasoning domains, we find that extensive continual pretraining on mathematical texts can improve performance on most non-MPS reasoning tasks generally. However, other dominant approaches of enhancing MPS performance fail to achieve significant gains on broad reasoning tasks. These findings indicate that most readily available data sources do not support the "math for AI" objective in enhancing non-MPS tasks. Identifying which data sources best contribute to the acquisition of complex reasoning skills remains a crucial question for future research.

## 1 INTRODUCTION

Cognitive neuroscience research has consistently demonstrated that learning to solve mathematical problems enhances general reasoning abilities in humans, as engaging in mathematical problem-solving promotes logical thinking, abstract reasoning, and transferable problem-solving strategies across various domains (Dehaene et al., 2004; Hawes & Ansari, 2020). This notion – that learning math fosters the development of general reasoning skills – points toward a "*math for AI*" vision, where incorporating mathematical reasoning data into AI training could help large language models (LLMs) develop more complex and versatile reasoning abilities. The "math for AI" goal is particularly relevant to recent attentions to complex reasoning abilities of LLMs (OpenAI, 2024), as mathematical problem-solving (MPS) is one of the few domains where large volumes of long and intricate CoT data can be generated or synthesized (Tang et al., 2024; Lu et al., 2024), making it a valuable data source to potentially learn complex reasoning. However, while numerous models have been developed to tackle mathematical problem-solving (Cobbe et al., 2021b; Yu et al., 2023; Luo et al., 2023a), their evaluations focus narrowly on benchmarks like GSM8K (Cobbe et al., 2021a) and MATH (Hendrycks et al., 2021b), and it is unclear whether these approaches and the accompanied datasets can really help learn other types of reasoning. Therefore, these works, whether intentional or not, fall within the "AI for math" scope and fail to demonstrate their impact for the "math for AI" objective. Thus, a key question remains: *Does learning mathematical problem-solving contribute*

*to the development of a model's general reasoning abilities, or does it merely enhance performance on MPS benchmarks?*

In this study, we conduct empirical analysis focusing on this central question. Specifically, we explore whether training LLMs on mathematical problem-solving tasks can help broader reasoning tasks beyond mathematics. We first identify three common training strategies to enhance LLMs' capabilities in solving mathematical problems: (1) *Continual pretraining on mathematical text* involves extending the pretraining of LLMs on large-scale mathematical text to enhance their adaptability to the mathematical domain, such as RhO-Math (Lin et al., 2024) and Deepseek-Math (Shao et al., 2024). (2) *Instruction pretraining on diverse QA pairs* is a method focused on training models using diverse question-answer pairs from raw texts, typically encompassing various formats and types of math problems (Yue et al., 2024; Cheng et al., 2024). (3) *Instruction tuning on MPS datasets* involves fine-tuning models on MPS datasets. This is the most common method adopted to learn mathematical problem-solving and lead to state-of-the-art performance (Yu et al., 2023; Gou et al., 2023; LI et al., 2024; Tong et al., 2024).

We perform control experiments and evaluate a series of model created by the three training strategies above, where the models are either from open-source checkpoints or our own training. We assess these models across multiple benchmarks involving MPS benchmarks and six types of non-MPS reasoning: mathematical reasoning (excluding problem-solving) , STEM reasoning, logical reasoning, commonsense reasoning, symbolic reasoning, and agent reasoning. When trained exclusively on mathematical texts, we observed that models tend to lose their ability to follow general instructions and become limited to performing only math-related tasks. To mitigate this effect, we also incorporated general chat-based data into the training process. This approach simulates a realistic development scenario where math-related training is integrated as part of broader model training, rather than isolating it to create a model solely capable of MPS tasks.

Our experimental results reveal that continual pretraining on raw mathematical texts enhances performance across a broader range of reasoning tasks. However, as we transition from continual pretraining to instruction pretraining and instruction tuning, the diversity of data drops, leading to decreased improvements. Particularly, MPS-oriented training negatively impacts performance on non-mathematical tasks. These findings also suggest that most open-source datasets in the math domain, which specifically target mathematical problem-solving, are unable to facilitate broader types of reasoning tasks to fulfill the "math for AI" goal. We encourage future research to reconsider the objectives when studying mathematical reasoning. If the goal is to enhance general reasoning capabilities rather than "AI for math", it may be worthwhile to explore which data sources, whether math-related or otherwise, can effectively contribute to the acquisition of more diverse reasoning skills.

In the final part of this work, we perform a pilot study, trying to identify potential data sources that could enhance reasoning skills. To this end, we experiment with three popular non-MPS SFT datasets that cover various thought reasoning processes, including coding-related tasks, a broad array of reasoning-intensive tasks and state-of-the-art conversational datasets. Unfortunately, none of these datasets demonstrated significant improvements across a wide spectrum of reasoning tasks. This points to a pessimistic conclusion that, in comparison to the extensive data used in pretraining, the relatively modest volume of SFT data is insufficient to substantially improve the model's general reasoning capabilities, even when the data originates from diverse domains.

## 2 METHODS

### 2.1 TRAINING PARADIGMS FOR MATHEMATICAL PROBLEM-SOLVING

The improvement of mathematical problem-solving abilities in LLMs has been explored through various training approaches, each with its own strengths and focus. Starting from a pretrained base model, in this study, we explore three prominent training strategies as followed. Due to the expensive cost of running some of the training paradigms, we obtain the required model from either the open-source checkpoints or our own training as we also detail next.

**Continual Pretraining on Mathematical Text.** In mathematics, where texts often involve multi-step reasoning and formal expressions, this approach helps models better grasp the reasoning pat-

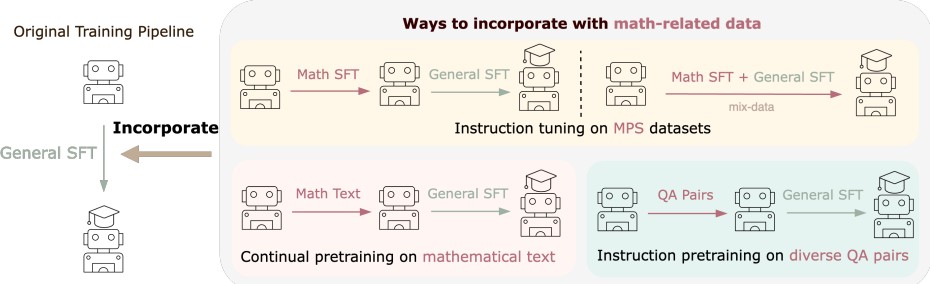

Figure 1: Three ways to incorporate math-related data into original training pipeline through hybrid training process. Original training pipeline is to SFT models with general conversation data. For the instruction tuning on MPS datasets, we conducted both two-stage training and mix-data training, for continual pretraining on mathematical text and instruction pretraining on diverse QA pairs, we only conducted the two-stage training.

terns (Lewkowycz et al., 2022). Due to the expensive cost of running continual pretraining, in this study, we experiment with two open-weight LLMs continually pretrained on mathematical-related text: RhO-Math (Lin et al., 2024) and DeepSeekMath (Shao et al., 2024). DeepSeekMath-Base is continual pretrained based on the DeepSeek-Coder-Base model using a large mathematical corpus called DeepSeekMath Corpus. It achieves 64.2% on GSM8K and 36.2% on the competition-level MATH dataset. Rho-Math-7B is continual pretraining with Selective Language Modeling method through OpenWebMath corpus on Mistral-7B, achieving 66.9% on GSM8K and 31.0% on MATH dataset. Distinct from normal continual pretraining, Rho-Math utilizes another reference model to select tokens and only optimize losses on the selected tokens. However, the reference model is created by training on task-specific SFT datasets. While Rho-Math demonstrated superior performance on mathematical problem-solving, in §3.3 we will show that this training scheme may potentially overfit on benchmark tasks as well, and fail to achieve significant gains on non-MPS tasks.

**Instruction Pretraining on Diverse QA Pairs.** Instruction pretraining using diverse question-answer (QA) pairs improves a model's generalization across diverse tasks while enhancing its instruction-following capabilities (Yue et al., 2024; Chung et al., 2024; Cheng et al., 2024). This approach involves with large QA datasets, often synthesized from raw text, encompassing various formats, complexities, and problem types. Typically, powerful LLMs like GPT-4 are used to filter raw text and generate relevant QA pairs. In our study, we leverage the open-weight MammoTH2 model (Yue et al., 2024) to evaluate it on broader tasks. MammoTH2 was trained on approximately 10 million QA pairs synthesized through open-source LLMs from a wide range of mathematical, science and engineering texts.

**Instruction Tuning on MPS Datasets.** Unlike continual pretraining or instruction pretraining on diverse QA pairs, this approach focuses on smaller, domain-specific datasets typically aligned with benchmark tasks. This is the most commonly used approach to boost MPS scores due to its efficiency. To assess whether models finetuned on MPS datasets can generalize beyond their source tasks, we use two different MPS-oriented datasets to train two models on our own : Math-COT SFT and Math-POT SFT. Math-COT SFT was trained on the MetaMath dataset (Yu et al., 2023), which draws primarily from the GSM8K and MATH benchmarks, all structured in a chain-of-thought (CoT) format. Math-POT SFT, on the other hand, was trained on the NuminaMath-TIR dataset (LI et al., 2024), which includes problems from GSM8K and MATH, as well as other benchmarks, with tasks presented in natural language and solutions in code snippets. The NuminaMath-TIR dataset directly leads to the NuminaMath model that wins a recent AI for Math competition.[1]

## 2.2 Hybrid Training

The training strategies described in §2.1, if exclusively used, could lead to the development of models specialized solely in mathematical reasoning tasks. However, this work focuses on studying "math for AI", the impact of math-related training and data on general model development. And it is a common practice to mix different sources of datasets to perform training (Xu et al., 2023;

---

[1]https://www.kaggle.com/competitions/ai-mathematical-olympiad-prize/leaderboard

Meta, 2024). Given this context, it is crucial for developers to understand: how would incorporating additional math-related training impact the original general training performance? To investigate this, we design our experiments to mimic the realistic setting, focusing on a simple yet prevalent training pipeline: a pretrained base model followed by the original SFT training (e.g., on general conversational data). We then conduct controlled experiments to introduce additional math-related data into this training pipeline, aiming to evaluate its influence on the model's performance across various tasks. we explore two different ways of integrating math-related training: two-stage training and mix-data training, as we detail below. The process is illustrated in Figure 1.

**Two-stage Training**   Since continual pretraining and instruction pretraining typically serve as an intermediate stage to obtain an enhanced base model followed by SFT training (Shao et al., 2024; Yue et al., 2024), we examine a two-stage training approach that injects math-related data in a mid-training stage. Specifically, in the first stage, one of the three methods outlined in §2.1 is applied, designed to strengthen the model's foundational mathematical reasoning abilities. In the second stage, we fine-tune these first-stage models using general conversation data to broaden their applicability to a variety of reasoning tasks, we choose UltraChat (Ding et al., 2023) as the general SFT dataset in this work, which is commonly used to create chat models (Tunstall et al., 2023). This process helps the models adapt to instruction-following tasks, thereby improving their versatility across different domains.

**Mix-data Training**   Considering that the two-stage training method may weaken a model's generalization ability due to catastrophic forgetting, we explore another commonly adopted training strategy for incorporating additional SFT datasets, which mixes various SFT data sources together. We only experiment this method for instruction tuning on MPS datasets, since the other two are designed to be conducted in a separate, intermediate training stage. In this mix-data training approach, the training data is a mixture of either Math-COT SFT or Math-POT SFT data combined with UltraChat data. Unlike two-stage training, where the model undergoes independent two sequential fine-tuning stages, the mix-data approach consolidates the training process into a single stage.

## 3 EXPERIMENTS

We consider seven particular models from three training strategies which aimed at enhancing the math reasoning capabilities. And we assess the generalization capabilities across multiple types of reasoning benchmarks of these models, encompassing both MPS and non-MPS tasks.

### 3.1 TRAINING SETUP

**Two-stage training setup**   We compare several models across the three studied training strategies to evaluate their performance on reasoning tasks. The models used in the first stage of training come from approaches in §2.1, which are outlined as follows:(1) For continual pretraining on mathematical text, we leveraged two existing checkpoints: `deepseek-math-7b-base` and `rho-math-7b-v0.1`. Their corresponding base models, are Deepseek-Coder-Base and Mistral-7B, respectively. (2) For instruction pretraining on diverse QA pairs , we used the checkpoint `MAmmoTH2-7B`, and Mistral-7B serves as its base model. (3) For instruction tuning on MPS datasets, we fine-tuned the base model `mistral-7b-v0.1` ourselves using the MetaMath (Yu et al., 2023) and NuminaMath-TIR (LI et al., 2024) datasets to get the Math-COT SFT model and the Math-POT SFT model. These models serve as the first-stage models for further tuning. After obtaining these first-stage models from each of three approaches, we performed a second-stage fine-tuning on both the math-specialized models and their corresponding base models. In this stage, we fine-tuned the models using the filtered UltraChat (Ding et al., 2023) data, which consists of general conversational content with approximately 200K samples.

**Mix-data training setup**   Additionally, we conducted mix-data training through these SFT datasets. The UltraChat data was combined with either MetaMath or NuminaMath-TIR data, randomly shuffled and mixed together. Then we fine-tuned the checkpoint `mistral-7b-v0.1` on these two mixture data. All the training methods that we study are summarized in Table 1.

Table 1: Models trained through two-stage training and mix-data training process. The baseline of DeepSeekMath (2-stage) is DeepSeek-Coder (2-stage), which is Deepseek-Coder-Base after Ultra-Chat tuning, while other final models' baseline is Mistral-7B (2-stage), which is Mistral-7B after UltraChat tuning.

| Model Training Process |
| --- |
| **Two-stage Training Process** |
| DeepSeek-Coder-Base → DeepSeekMath Corpus → DeepSeekMath-Base → UltraChat → DeepSeekMath (2-stage) |
| Mistral-7B-Base → OpenWebMath Corpus → Rho-Math-7B → UltraChat → Rho-Math-7B (2-stage) |
| Mistral-7B-Base → WebInstruct → MAmmoTH2-7B → UltraChat → MAmmoTH2-7B (2-stage) |
| Mistral-7B-Base → MetaMath → Math-COT SFT → UltraChat → Math-COT SFT (2-stage) |
| Mistral-7B-Base → NuminaMath-TIR → Math-POT SFT → UltraChat → Math-POT SFT (2-stage) |
| **Mix-data Training Process** |
| Mistral-7B-Base → MetaMath + UltraChat → Math-COT SFT (mixed) |
| Mistral-7B-Base → NuminaMath-TIR + UltraChat → Math-POT SFT (mixed) |

Table 2: Benchmarks in Each Reasoning Domain.

| Reasoning Domain | Benchmarks |
| --- | --- |
| Math Reasoning (problem-solving) | GSM8K, GSM8K MQA, MATH, MMLU-math |
| Math Reasoning (excluding problem-solving) | MR-BEN-math, DocMath (Zhao et al., 2024) |
| Logical Reasoning | ZebraLogic (Bill Yuchen Lin, 2024), ProofWriter (Tafjord et al., 2020), LogiQA (Liu et al., 2020) |
| STEM Reasoning | GPQA (Rein et al., 2023), MMLU-stem |
| Commonsense Reasoning | NQ (Lee et al., 2019), SWAG (Zellers et al., 2018), WinoGrande (Sakaguchi et al., 2021), ARC-challenge (Clark et al., 2018) |
| Symbolic Reasoning | BBH (Suzgun et al., 2022) |
| Agent Reasoning | MiniWoB++ (Liu et al., 2018) |

We use the sanitized version of Ultrachat provided by HuggingFace[2], To balance the exposure of the math and general conversation data, we randomly selected 200K data samples from MetaMath for SFT. For NumniaMath-TIR only has 72K items, so we keep all the samples for SFT. More training hyperparameters are showed in Appendix C.1.

## 3.2 EVALUATION DATASETS

To evaluate models' multi-dimensional reasoning capabilities, we choose seven reasoning tasks: math reasoning (problem-solving) (MPS), math reasoning (exculding problem-solving), logical reasoning, STEM reasoning, commonsense reasoning, symbolic reasoning and agent reasoning. The corresponding benchmarks are shown in Table 2. The GSM8K MQA dataset is derived from the original GSM8K format, repurposed into a multiple-choice question format. The MMLU-math and MMLU-stem are the math and stem sub-categories of MMLU (Hendrycks et al., 2021a). The MR-BEN-math is only the math subject of MR-BEN (Zeng et al., 2024). See more introduction of benchmarks in Appendix C.3

---

[2]https://huggingface.co/datasets/HuggingFaceH4/ultrachat_200k

Table 3: Performance of different models. Models are after the two-stage training or mix-data training. Absolute accuracy changes compared to the baselines are highlighted. The baseline of DeepSeekMath (2-stage) is the DeepSeek-Coder (2-stage). The baseline of other models are the Mistral-7B (2-stage). Both baselines are finetuned with UltraChat data. MPS: Math (problem-solving). MR: Math (excluding problem-solving). CS: Commonsense.

| Model | Math Resaoning | | Non-Math Resaoning | | | | |
|---|---|---|---|---|---|---|---|
| | MPS | MR | Logical | STEM | CS | Symbolic | Agent |
| Mistral-7B (2-stage) | 38.8 | 16.4 | 22.3 | 39.6 | 53.6 | 55.3 | 50.4 |
| DeepSeek-Coder (2-stage) | 43.5 | 25.1 | 21.6 | 37.3 | 42.8 | 56.8 | 57.6 |
| **(1) Continual pretraining on raw text** | | | | | | | |
| DeepSeekMath (2-stage) | 57.7 ↑14.2 | 26.4 ↑1.3 | 21.3 ↓0.3 | 41.9 ↑4.6 | 43.6 ↑0.8 | 60.6 ↑3.8 | 45.9 ↓11.7 |
| Rho-Math-7B (2-stage) | 54.0 ↑15.2 | 19.3 ↑2.9 | 21.9 ↓0.4 | 40.8 ↑1.2 | 49.4 ↓4.2 | 57.0 ↑1.7 | 50.3 ↓0.1 |
| **(2) Instruction pretraining on large-scale diverse QA pairs** | | | | | | | |
| MAmmoTH2-7B (2-stage) | 56.0 ↑17.2 | 21.4 ↑5.0 | 23.7 ↑1.4 | 41.0 ↑1.4 | 51.5 ↑2.1 | 56.4 ↑1.1 | 50.3 ↓0.1 |
| **(3) Instruction tuning on MPS datasets** | | | | | | | |
| Math-COT SFT (2-stage) | 44.6 ↑5.8 | 18.1 ↑1.7 | 22.9 ↑0.6 | 37.7 ↓1.9 | 53.5 ↓0.1 | 53.8 ↓1.5 | 50.4 |
| Math-POT SFT (2-stage) | 42.0 ↑3.2 | 18.1 ↑1.7 | 22.2 ↓0.1 | 39.5 ↓0.1 | 53.5 ↓0.1 | 54.1 ↓1.2 | 45.4 ↓5.0 |
| Math-COT SFT (mixed) | 54.4 ↑15.6 | 20.1 ↑3.7 | 22.4 ↑0.1 | 38.1 ↓1.5 | 52.5 ↓1.1 | 49.5 ↓5.8 | 52.1 ↑1.7 |
| Math-POT SFT (mixed) | 52.0 ↑13.2 | 20.4 ↑4.0 | 22.5 ↑0.2 | 39.0 ↓0.6 | 52.5 ↓1.1 | 52.8 ↓2.5 | 57.7 ↑7.3 |

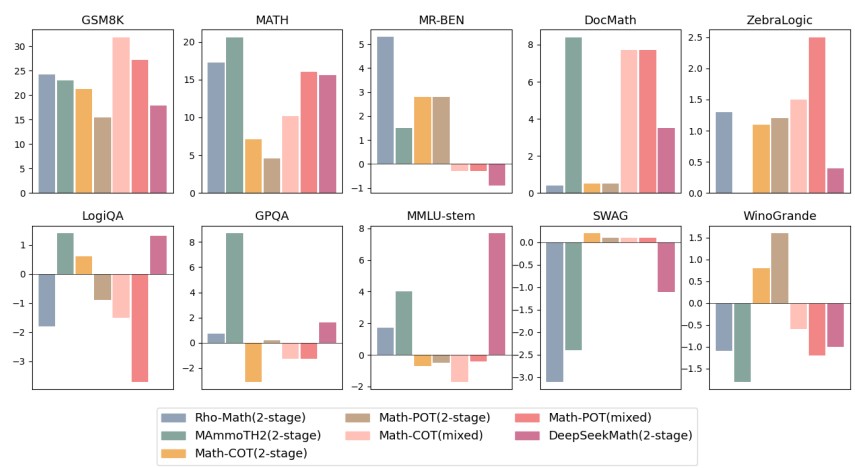

Figure 2: Relative change across specific benchmarks for math-related models after two-stage training or mix-data training process.

## 3.3 MAIN RESULTS

Table 3 presents the performance of models of three kinds of training strategies on seven kinds of reasoning tasks. The results are calculated as the average value across each reasoning domain. Results for each benchmark are showed in Appendix A.1.

**Learning mathematical problem-solving helps mathematical reasoning in general** We could observe that all models demonstrate improvements on math problem-solving (MPS) tasks, but the gains for Math-COT and Math-POT models with two-stage training are relatively smaller compared to other methods. Mixed training for Math-COT and Math-POT lead to much higher performance on MPS tasks. On other types of math reasoning tasks which are not problem-solving, all models demonstrate gains despite smaller magnitudes compared to problem-solving tasks. This suggests

that learning mathematical problem-solving is able to generalize and help other types of mathematical reasoning as well.

**Continual pretraining generally improves non-mathematical reasoning while selective continual pretraining falls short**   The improvements on mathematical reasoning tasks are actually expected, yet we note that this work emphasizes more the effect on other non-mathematical reasoning tasks. We first observe that continual pretraining of DeepSeekMath enhances performance in 3 out of 5 non-mathematical tasks, achieving a notable increase of 4.6 points in STEM reasoning and 3.8 points in symbolic reasoning. DeepSeekMath is also the only one among these models that can achieve an average of over 2-point gain on some non-mathematical reasoning domains. Conversely, Rho-Math, another variant of continual pretraining, only showed improvements in 2 out of 5 non-mathematical reasoning domains with limited gains under 2 points. In more detail, as shown in Figure 2, the Rho-Math perform worse than DeepSeekMath on more datasets. As introduced in §2.1, Rho-Math employs a selective language modeling loss that leverages a reference model to help select tokens for optimization – this reference model, trained on task-specific SFT datasets, may introduce biases that compromise the generalization capacity. Previously, the extent of this compromise was unknown as only mathematical problem-solving tasks were assessed. Therefore, we urge the research community to conduct to more comprehensive evaluations of a model's reasoning capabilities, to gain a more complete understanding of different training algorithms. Otherwise, in the case of Rho-Math, although it achieves similar gains on MPS benchmarks as DeepSeekMath while being trained on far fewer tokens, the trade-offs compared to standard continual pretraining were not initially clear, as we now demonstrate.

**Instruction pretraining sometimes help non-mathematical reasoning, while instruction tuning generally impairs**   We observe that instruction pretraining with the MAmmoTH2 model improves 3 out of 5 non-mathematical reasoning tasks, despite small gains around 1 point. However, instruction tuning on MPS datasets, the most commonly adopted method to learn mathematical problem solving, undermines the original training pipeline on most non-mathematical reasoning tasks, except for the agent reasoning task. This points to a pessimistic reality: most previous efforts that develop new MPS datasets and advance state-of-the-art for mathematical reasoning may not generalize to facilitate learning in other types of reasoning. In fact, the created data resources may even negatively impact other reasoning abilities, a phenomenon that contradicts intuitive expectations based on human learning studies.

**Agent task specicfic tuning**   As the models exhibit significant variation in performance on the agent reasoning task, which is likely due to the fixed-format code required as input for agent tasks. The performance comparison becomes highly dependent on the models' ability to generate accurate code. To reduce this disparity, we replaced the second-stage UltraChat data with task-specific data related to the benchmark. Specifically, we used data from Mini-Wob++, generated by Claude-2, as the second-stage training data. The results of this adjustment are shown in Figure 3. We observe that Rho-Math, MAmmoTH2 and DeepSeekMath all demonstrate improvement over the base model, while Math-COT SFT and Math-POT SFT continue to underperform, reinforcing the notion that models trained via SFT have limited generalization capabilities.

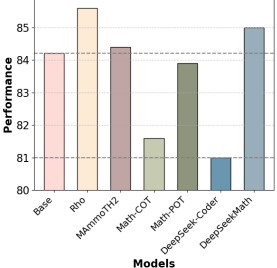

Figure 3: Performance on MiniWob++ for models tuning on task specific data.

## 4 WHAT OTHER DATA SOURCES CONTRIBUTE TO REASONING – A PILOT STUDY

So far, we have explored the effect of various math-related data sources on general reasoning learning, and we have concluded that only continual training with raw math text has a significantly positive effect on general reasoning learning. However, continual pretraining is typically large-scale and computationally expensive. In this section, we perform a pilot study to search for efficient SFT datasets from non-mathematical tasks, to examine whether they can help learn reasoning. Specifically, we identify the following three non-MPS SFT datasets as our targets to study, based on their diverse task coverage as showed in Table 4:

Table 4: Areas covered by the three selected non-MPS SFT datasets.

| Dataset | Size | Code Algorithm | Reasoning | General Knowledge |
|---|---|---|---|---|
| Magicoder-Evol-Instruct | 110K | ✓ | ✗ | ✗ |
| Magpie-Reasoning | 150K | ✓ | ✓ | ✗ |
| OpenOrca | 200K | ✓ | ✓ | ✓ |

Table 5: Performance of mix-data training models of non-MPS data on reasoning tasks. Base model is Mistral-7B after UltraChat tuning. Absolute accuracy changes are highlighted. MPS: Math (problem-solving). MR: Math (excluding problem-solving). CS: Commonsense. Results are averaged across each reasoning domain.

| | Math Resaoning | | Non-Math Resaoning | | | | |
|---|---|---|---|---|---|---|---|
| Model | MPS | MR | Logical | STEM | CS | Symbolic | Agent |
| Mistral-7B (2-stage) | 38.8 | 16.4 | 22.3 | 39.6 | 53.6 | 55.3 | 50.4 |
| **Mix-data training on non-MPS datasets** | | | | | | | |
| Magicoder-Evol-Instruct SFT (mixed) | 38.1 ↓0.7 | 20.8 ↑4.4 | 23.5 ↑1.2 | 36.7 ↓2.9 | 52.9 ↓0.7 | 52.9 ↓2.4 | 49.0 ↓1.4 |
| Magpie-Reasoning SFT (mixed) | 47.0 ↑8.2 | 21.1 ↑4.7 | 21.8 ↓0.5 | 38.8 ↓0.8 | 52.8 ↓0.8 | 50.9 ↓4.4 | 54.5 ↑4.1 |
| OpenOrca SFT (mixed) | 30.4 ↓8.4 | 24.8 ↑8.4 | 20.6 ↓1.7 | 40.1 ↑0.5 | 53.1 ↓0.5 | 54.6 ↓0.7 | 56.5 ↑6.1 |

- *Magicoder-Evol-Instruct*[3] (Wei et al., 2023) is used primarily to enhance code generation capabilities in LLMs. The dataset was decontaminated and repurposed from an earlier open-source instruction dataset, Evol-CodeAlpaca[4], which has augmented questions and answers by GPT-4. The dataset helping improve the performance of LLMs on code generation and program algorithm tasks, particularly in diverse programming contexts.

- *Magpie-Reasoning*[5] is a specialized SFT dataset designed to improve the reasoning capabilities of LLMs. It is generated by Qwen2-72B-Instruct (Yang et al., 2024) and Llama-3-70B Instruct (Meta, 2024) using Magpie (Xu et al., 2024b). It consists of 150K samples of conversations, covering a mix of tasks including mathematical reasoning, code-based reasoning, and general logic-based problem-solving.

- *OpenOrca*[6] is a large, open-domain dataset that spans diverse fields, including math, science, general knowledge, and other multi-domain tasks, with the distributions outlined in Orca (Mukherjee et al., 2023). This dataset is augmented from FLAN collection data (Longpre et al., 2023) with GPT-4. Given resource limitations, we performed SFT on 200K samples.

**Setup** Based on the results in §3.3, we found that, when using the same mathematical SFT datasets, mix-data training model generally outperforms two-stage training model in terms of performance. Therefore, we employ mix-data training process for the three non-MPS datasets in this section. Specifically, each non-MPS SFT dataset is randomly mixed with filtered UltraChat data, following the same approach used in §3.1. The initial model remains `mistral-7b-v0.1`.

**Results** As shown in Table 5, despite incorporating diverse datasets such as Magicoder-Evol-Instruct, Magpie-Reasoning, and OpenOrca, the generalization capability of the models across different reasoning tasks remains limited. See detail results on all benchmarks in Appendix A.1. Additionally, the performance among each model still remains some distinction. The Magicoder-Evol-Instruct SFT model shows improvements in fewer areas compared to the other models. This may be attributed to the narrower scope of this dataset, which primarily focuses on code-related tasks. In contrast, the Magpie-Reasoning SFT model demonstrates performance improvements in a

---

[3]https://huggingface.co/datasets/ise-uiuc/Magicoder-Evol-Instruct-110K

[4]https://huggingface.co/datasets/theblackcat102/evol-codealpaca-v1

[5]https://huggingface.co/datasets/Magpie-Align/Magpie-Reasoning-150K

[6]https://huggingface.co/datasets/Open-Orca/OpenOrca

broader range of tasks. This is likely due to its more balanced dataset, which covers both code algorithms and reasoning tasks. Interestingly, the OpenOrca SFT model, despite its broader coverage of reasoning, coding, and general knowledge, shows relatively fewer performance gains compared to Magpie. This could be due to the complexity and diversity of the OpenOrca dataset, which might introduce competing learning objectives, causing the model to struggle in balancing between different types of tasks. While there are some localized improvements in certain domains, such as agent reasoning, where the models exhibit noticeable gains, the overall trend indicates that SFT method, even with diverse and extensive datasets, struggles to generalize effectively across a wide range of reasoning challenges. How to find efficient datasets to enhance general reasoning abilities of LLMs still remain as a critical challenge for future researches to study.

## 5 RELATED WORKS

While LLMs exhibit remarkable performance out of the box, especially in tasks that require pattern recognition and language understanding (Zhao et al., 2022; Brown, 2020; Wei et al., 2022; Creswell et al., 2022), their ability to perform complex reasoning often requires additional refinement through targeted training methods.

**Supervised Fine-Tuning**   A key method for enhancing LLM performance is Supervised Fine-Tuning (SFT). SFT not only improves a model's ability to follow instructions but also enhances its performance on intricate tasks requiring specialized knowledge by training on well-curated datasets (Xu et al., 2023; Zhou et al., 2023; Wu et al., 2023b; Yuan et al., 2023b; Chen et al., 2023b). As LLMs continue to evolve, researchers also employ SFT as a crucial step in tailoring the models for more complex reasoning scenarios or tasks (Huang & Chang, 2022; Wang et al., 2023b). In the context of mathematical reasoning, SFT has demonstrated substantial improvements in model performance (Cobbe et al., 2021b; Nye et al., 2021; Yuan et al., 2023a; Yue et al., 2023; Wang et al., 2023a; Li et al., 2023; Liu et al., 2023; Chen et al., 2024). For instance, the MetaMath model, fine-tuned on an augmented GSM8K and MATH dataset, demonstrated notable improvements on mathematical problem-solving benchmarks (Yu et al., 2023). In addition to mathematical reasoning, SFT has also been utilized to achieve better results on other types of reasoning tasks. It has been applied to domains like commonsense reasoning (Huang et al., 2022; Bian et al., 2024) and logical reasoning (Luo et al., 2023b; Chen et al., 2023c; Li et al., 2024), Moreover, researchers also reveal that SFT also helps LLMs handle more dynamic and context-rich tasks like agent-based reasoning (Gou et al., 2023; Chen et al., 2023a), where understanding interactions and goals in simulated environments is essential.

**Continual Pretrain**   Continual pretraining is another widely adopted approach to enhance the performance of LLMs in specific domains (Aharoni & Goldberg, 2020). Unlike SFT, which relies on task-specific datasets, continual pretraining exposes models to large-scale, domain-relevant corpora Paster et al. (2023); Wang et al. (2023c). The large-scale corpora expands the model's knowledge base and helps the model generalize better within specialized areas (Jin et al., 2021; Gupta et al., 2023; Ke et al., 2023; Wu et al., 2023a; Bian et al., 2024). In the realm of mathematical problem solving, continual pretraining also has been instrumental in improving models' abilities to tackle complex reasoning tasks (Lewkowycz et al., 2022; Lin et al., 2024; Shao et al., 2024).

## 6 CONCLUSION

In this paper, we explored the generalization potential of three different training strategies to learn mathematical problem-solving. Our experiments evaluated models trained using (1) continual pretraining on mathematical text, (2) instruction tuning on diverse QA pairs, and (3) instruction tuning on MPS datasets. The results indicate that only continual pretraining on raw mathematical text can lead to significant gains on most domains. In contrast, models fine-tuned on MPS SFT datasets struggled to generalize beyond math-specific tasks and even impaired other reasoning abilities. These observations imply that previous researches on mathematical reasoning may put too much focus on mathematical problem-solving task, which stay far away from the "math for AI" goal. Future research could explore how both math-related or non-math datasets can be leveraged to better develop models capable of handling a wider variety of reasoning tasks.

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

APPENDIX

# A  RESULT DETAILS

## A.1  BENCHMARK RESULTS

In this section, we present the detailed results of each trained model. Table 6 provides detailed results for each benchmark of math (problem-solving) and STEM reasoning tasks of math-related models after two-stage training or mix-data training. We could observe the Math-SFT (2-stage) models have some decline in certain MPS benchmarks, indicate that these two models even have the limitation on math problem solving question that they are not familiar with. Besides, most of the SFT models not performed well on the STEM reasoning tasks, while the continual pretrained models, Rho-Math and DeepSeekMath showed enhancement. Table 7 presents detailed results for each benchmark of math (excluding problem-solving) and logical reasoning of math-related models after two-stage training or mix-data training. MAmmoTH2 and Math-COT SFT (2-stage) model showed consistent improvement across these benchmarks, while other models fail to demonstrate superior performance especially in logical reasoning benchmarks. Table 8 shows the commonsense benchmarks results of 2-stage models and mix-data training models. We could observe that models not outperform in this reasoning domain, particular both continual pretrained and instruction pretrained models. This may suggest that when models incorporate math-related data in training process, it potentially shifts the focus away from the general commonsense reasoning patterns.

Table 9 provides detailed results for each benchmark of math (problem-solving) and logical reasoning tasks of models SFT through mix-data training process on non-MPS datasets. We could observe that all models indicate a drop on the benchmarks expect for GSM8K. This indicate this non-MPS data do not enhance the models capbility on math problem solving. Table 10 presents detailed results for each benchmark of math (excluding problem-solving) and logical reasoning of models SFT through mix-data training process on non-MPS datasets. These models demonstrate similar proficiency across these benchmarks. Table 11 presents detailed results for each benchmark of commonsense reasoning of models SFT through mix-data training process on non-MPS datasets. Same to the math-related models, these models show a reduction among commonsense reasoning task. This may indicate that these non-MPS data also not help to develop the necessary capabilities of models for solving general commonsense reasoning problems.

Table 6: Detailed results on math (problem-solving) and STEM reasoning benchmarks of two-stage training and mix-data training models. Absolute accuracy changes compared to the baselines are highlighted.

| Model | Math Reasoning (problem-solving) | | | | STEM Reasoning | |
|---|---|---|---|---|---|---|
| | GSM8K | GSM8K MQA | MATH | MMLU-math | GPQA | MMLU-stem |
| Mistral-7B (2-stage) | 40.6 | 56.9 | 12.3 | 45.5 | 30.8 | 48.4 |
| DeepSeek-Coder (2-stage) | 48.8 | 55.1 | 18.9 | 51.1 | 28.8 | 45.7 |
| **(1) Continual pretraining on raw text** | | | | | | |
| DeepSeekMath (2-stage) | 66.7 ↑17.9 | 70.1 ↑15.0 | 34.5 ↑15.6 | 59.6 ↑8.5 | 30.4 ↑1.6 | 53.4 ↑7.7 |
| Rho-Math-7B (2-stage) | 64.8 ↑24.2 | 65.7 ↑8.8 | 29.6 ↑17.3 | 55.8 ↑10.3 | 31.5 ↑0.7 | 50.1 ↑1.7 |
| **(2) Intruction pretraining on diverse QA pairs** | | | | | | |
| MAmmoTH2-7B (2-stage) | 63.6 ↑23.0 | 72.2 ↑15.3 | 32.9 ↑20.6 | 55.3 ↑9.8 | 29.5 ↓1.3 | 52.4 ↑4.0 |
| **(3) Instruction tuning on MPS datasets** | | | | | | |
| Math-COT SFT (2-stage) | 61.8 ↑21.2 | 53.1 ↓3.8 | 19.4 ↑7.1 | 44.1 ↓1.4 | 27.7 ↓3.1 | 47.7 ↓0.7 |
| Math-POT SFT (2-stage) | 56.0 ↑15.4 | 52.2 ↓4.7 | 16.9 ↑4.6 | 43.0 ↓2.5 | 31.0 ↑0.2 | 47.9 ↓0.5 |
| Math-COT SFT (mixed) | 72.4 ↑31.8 | 74.7 ↑17.8 | 22.5 ↑10.2 | 48.0 ↑2.5 | 29.5 ↓1.3 | 46.7 ↓1.7 |
| Math-POT SFT (mixed) | 67.8 ↑27.2 | 65.9 ↑9.0 | 28.3 ↑16.0 | 45.9 ↑0.4 | 29.9 ↓0.9 | 48.0 ↓0.4 |

Table 7: Detailed results on math (excluding problem-solving) and logical reasoning benchmarks of two-stage training and mix-data training models. Absolute accuracy changes compared to the baselines are highlighted.

| | Math Reasoning (excluding problem-solving) | | Logical Reasoning | | |
| --- | --- | --- | --- | --- | --- |
| Model | MR-BEN-math | DocMath | ZebraLogic | LogiQA | ProofWriter |
| Mistral-7B (2-stage) | 21.5 | 11.3 | 4.8 | 29.5 | 32.5 |
| DeepSeek-Coder (2-stage) | 35.2 | 15.0 | 4.7 | 25.4 | 34.8 |
| **(1) Continual pretraining on raw text** | | | | | |
| DeepSeekMath (2-stage) | 34.3 $\downarrow$0.9 | 18.5 $\uparrow$3.5 | 5.1 $\uparrow$0.4 | 26.7 $\uparrow$1.3 | 32.2 $\downarrow$2.6 |
| Rho-Math-7B (2-stage) | 26.8 $\uparrow$5.3 | 11.7 $\uparrow$0.4 | 6.1 $\uparrow$1.3 | 27.7 $\downarrow$1.8 | 32.0 $\downarrow$0.5 |
| **(2) Intruction pretraining on diverse QA pairs** | | | | | |
| MAmmoTH2-7B (2-stage) | 23.0 $\uparrow$1.5 | 19.7 $\uparrow$8.4 | 4.8 | 30.9 $\uparrow$1.4 | 35.5 $\uparrow$3.0 |
| **(3) Instruction tuning on MPS datasets** | | | | | |
| Math-COT SFT (2-stage) | 24.3 $\uparrow$2.8 | 11.8 $\uparrow$0.5 | 5.9 $\uparrow$1.1 | 30.1 $\uparrow$0.6 | 32.7 $\uparrow$0.2 |
| Math-POT SFT (2-stage) | 24.3 $\uparrow$2.8 | 11.8 $\uparrow$0.5 | 6.0 $\uparrow$1.2 | 28.6 $\downarrow$0.9 | 32.0 $\downarrow$0.5 |
| Math-COT SFT (mixed) | 21.2 $\downarrow$0.3 | 19.0 $\uparrow$7.7 | 6.3 $\uparrow$1.5 | 28.0 $\downarrow$1.5 | 32.8 $\uparrow$0.3 |
| Math-POT SFT (mixed) | 21.2 $\downarrow$0.3 | 19.5 $\uparrow$8.2 | 7.3 $\uparrow$2.5 | 25.8 $\downarrow$3.7 | 34.5 $\uparrow$2.0 |

Table 8: Detailed results on commonsense reasoning benchmarks of two-stage training and mix-data training models. Absolute accuracy changes compared to the baselines are highlighted.

| | Commonsense Reasoning | | | |
| --- | --- | --- | --- | --- |
| Model | NQ | SWAG | WinoGrande | ARC-challenge |
| Mistral-7B (2-stage) | 29.5 | 58.8 | 72.1 | 54.1 |
| DeepSeek-Coder (2-stage) | 13.7 | 52.7 | 64.5 | 40.1 |
| **(1) Continual pretraining on raw text** | | | | |
| DeepSeekMath (2-stage) | 13.0 $\downarrow$0.7 | 51.6 $\downarrow$7.2 | 63.5 $\downarrow$1.0 | 46.1 $\uparrow$5.0 |
| Rho-Math-7B (2-stage) | 21.0 $\downarrow$8.5 | 55.7 $\downarrow$3.1 | 71.0 $\downarrow$1.1 | 50.0 $\downarrow$4.1 |
| **(2) Intruction pretraining on diverse QA pairs** | | | | |
| MAmmoTH2-7B (2-stage) | 22.8 $\downarrow$6.7 | 56.4 $\downarrow$2.4 | 70.3 $\downarrow$1.8 | 56.5 $\uparrow$2.4 |
| **(3) Instruction tuning on MPS datasets** | | | | |
| Math-COT SFT (2-stage) | 29.5 | 59.0 $\uparrow$0.2 | 72.9 $\uparrow$0.8 | 52.7 $\downarrow$1.4 |
| Math-POT SFT (2-stage) | 29.0 $\downarrow$0.5 | 58.9 $\uparrow$0.1 | 73.7 $\uparrow$1.6 | 52.5 $\downarrow$1.6 |
| Math-COT SFT (mixed) | 27.0 $\downarrow$2.5 | 58.9 $\uparrow$0.1 | 71.5 $\downarrow$0.6 | 52.5 $\downarrow$1.6 |
| Math-POT SFT (mixed) | 26.7 $\downarrow$2.8 | 59.0 $\uparrow$0.2 | 70.9 $\downarrow$1.2 | 53.4 $\downarrow$0.7 |

Table 9: Detailed results on math (problem-solving) and STEM reasoning benchmarks of models with mix-data training process on non-MPS datasets (based on Mistral-7B). Absolute accuracy changes compared to the baselines are highlighted.

| | Math Reasoning (problem-solving) | | | | STEM Reasoning | |
| --- | --- | --- | --- | --- | --- | --- |
| Model | GSM8K | GSM8K MQA | MATH | MMLU-math | GPQA | MMLU-stem |
| Mistral-7B (2-stage) | 40.6 | 56.9 | 12.3 | 45.5 | 30.8 | 48.4 |
| **Mix-data training on non-MPS datasets** | | | | | | |
| Magicoder-Evol-Instruct SFT (mixed) | 43.1 $\uparrow$2.5 | 55.5 $\downarrow$1.4 | 10.9 $\downarrow$1.4 | 42.9 $\downarrow$2.6 | 26.1 $\downarrow$4.7 | 47.2 $\downarrow$1.2 |
| Magpie-Reasoning SFT (mixed) | 62.7 $\uparrow$22.1 | 65.6 $\uparrow$8.7 | 15.7 $\uparrow$3.4 | 44.0 $\downarrow$1.05 | 29.2 $\downarrow$1.6 | 48.4 |
| OpenOrca SFT (mixed) | 49.1 $\uparrow$8.5 | 21.8 $\downarrow$35.1 | 11.2 $\downarrow$1.1 | 39.4 $\downarrow$6.1 | 31.5 $\downarrow$0.7 | 48.7 $\uparrow$0.3 |

# B MORE RESULT ANALYSIS

Figure 4 illustrates the comparative performance between the first-stage models and the final models across multiple reasoning domains. From the radar chart, it is evident that the final models usually exhibit a consistent improvement. Additionally, for models tuned on MPS datasets, the mix-data training process showed slight improvements over the two-stage training method on certain

Table 10: Detailed results on math (exculding problem-solving) and logical reasoning benchmarks of models with mix-data training process on non-MPS datasets (based on Mistral-7B). Absolute accuracy changes compared to the baselines are highlighted.

| Model | Math Reasoning (excluding problem-solving) | | Logical Reasoning | | |
| | MR-BEN-math | DocMath | ZebraLogic | LogiQA | ProofWriter |
| --- | --- | --- | --- | --- | --- |
| Mistral-7B (2-stage) | 21.5 | 11.3 | 4.8 | 29.5 | 32.5 |
| **Mix-data training on non-MPS datasets** | | | | | |
| Magicoder-Evol-Instruct SFT (mixed) | 23.7 ↑2.2 | 17.8 ↑6.5 | 6.4 ↑1.6 | 29.3 ↓0.2 | 34.8 ↑2.3 |
| Magpie-Reasoning SFT (mixed) | 22.9 ↑1.4 | 19.2 ↑7.9 | 5.0 ↑0.2 | 29.0 ↓0.5 | 31.3 ↓1.2 |
| OpenOrca SFT (mixed) | 28.4 ↑6.9 | 21.2 ↑9.9 | 5.2 ↑0.3 | 26.0 ↓3.5 | 30.5 ↓2.0 |

Table 11: Detailed results on commonsense reasoning benchmarks of models with mix-data training process on non-MPS datasets (based on Mistral-7B). Absolute accuracy changes compared to the baselines are highlighted.

| Model | Commonsense Reasoning | | | |
| | NQ | SWAG | WinoGrande | ARC-challenge |
| --- | --- | --- | --- | --- |
| Mistral-7B (2-stage) | 29.5 | 58.8 | 72.1 | 54.1 |
| **Mix-data training on non-MPS datasets** | | | | |
| Magicoder-Evol-Instruct SFT (mixed) | 27.7 ↓1.8 | 59.2 ↑0.4 | 71.3 ↓0.8 | 53.3 ↓0.8 |
| Magpie-Reasoning SFT (mixed) | 27.2 ↓2.3 | 59.6 ↑0.8 | 71.1 ↓1.0 | 53.2 ↓0.9 |
| OpenOrca SFT (mixed) | 27.4 ↓2.1 | 59.6 ↑0.8 | 72.5 ↑0.4 | 52.7 ↓1.4 |

benchmarks. Figure 5 shows that the models with mix-data training have higher confidence than the models after UltraChat tuning. Even compare to the models after task specific data tuning, the mixed data models have more samples with higher confidence.

# C  EXPERIMENT DETAILS

## C.1  TRAINING HYPERPARAMETERS

The training process was carried out using the AdamW optimizer with a cosine learning rate scheduler. The training utilized a warmup ratio of 0.1 and set the batch size as 512. Additionally, the training was conducted using DeepSpeed with stage2 configuration. All of these SFT models were fine-tuned using the FastChat (Zheng et al., 2023) framework with a peak learning rate of 2e-5. Based on the FashChat original framework, we also adapted the sequence packing technique to speed up the training.

For the first stage models with instruction tuning on MPS datasets (Math-COT SFT and Math-POT SFT), we trained for 3 epochs with math-related data, and for the UltraChat tuning stage, we trained 1 epoch. For the mix-data training, we trained for 3 epochs for both Math-COT SFT (mixed) model and Math-POT SFT (mixed) model. Besides, when we replace the second stage data with task specific data for MiniWob++, we trained for 3 epochs to force the models learned the ability of generate the correct format code. We trained all models on a cluster with 8 NVIDIA A800 GPUs.

## C.2  EVALUATION DETAILS

For evaluation, we assessed the majority of datasets using the lm-evaluation-harness (Gao et al., 2024) framework. For other datasets that not be included in lm-evaluation-harness, we opted to use the original scripts provided with the datasets if existed. For the MiniWob++ task, we adapted the script from ENVISIONS (Xu et al., 2024a) and leverage the ChromeDriver to simulate the agent.

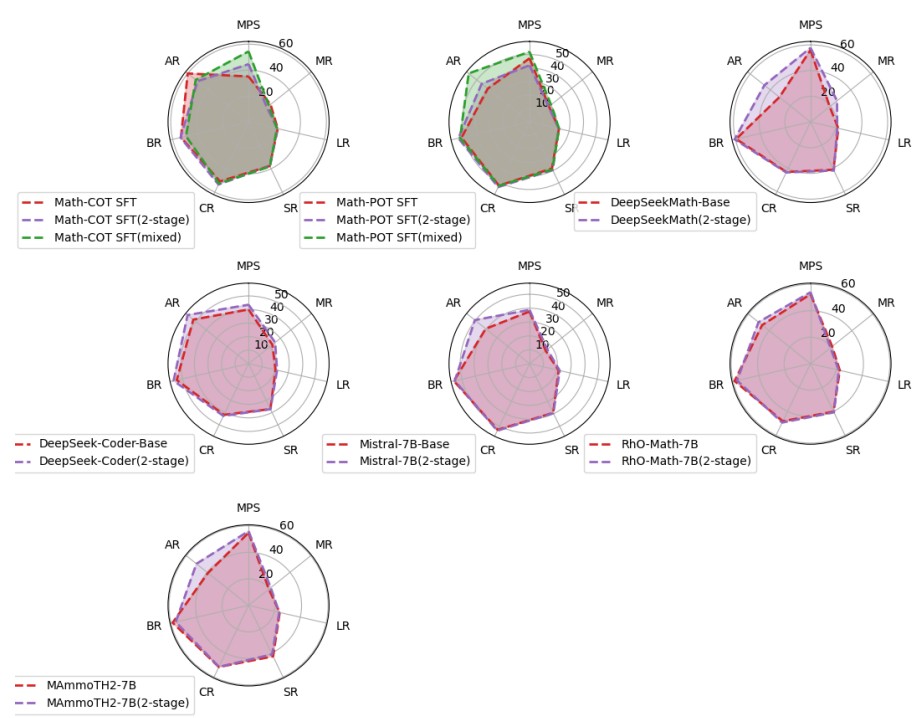

Figure 4: Performance for first stage models and final models after two-stage training or mix-data training. MPS: Math Reasoning (problem-solving). MR: Math Reasoning (excluding problem-solving). LR: Logical Reasoning. SR: STEM Reasoning. CR: Commonsense Reasoning. BR: Symbolic Reasoning. AR: Agent Reasoning.

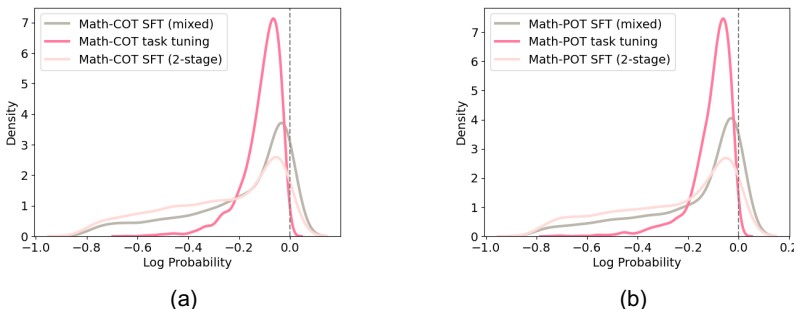

Figure 5: Density of log probability across various math SFT models on MiniWob++. Math SFT task tuning means the second stage tuning is through task specific data instead of UltraChat.

## C.3 BRIEF INTROCUCION OF BENCHMARKS

Here are the brief introduction to each benchmark. For some complex benchmarks, we also present the corresponding prompt for evaluation.

**GSM8K** GSM8K (Cobbe et al., 2021a) is a dataset specifically designed for evaluating LLMs in the domain of multi-step mathematical reasoning. The problem in this dataset are high quality linguistically diverse grade school math word problems created by human problem writers.

**GSM8K MQA** This is a dataset where we reformatted the original GSM8K dataset into multiple-choice questions. We kept the original question and let GPT-4o generate other three confusing answers based on the original answer. Models need to generate the option letter of the correct answer.

**MATH** MATH (Hendrycks et al., 2021b) test dataset contains 5,000 challenging competition mathematics problems. Each problem in MATH has a full step-by-step solution which can be used to teach models to generate answer derivations and explanations.

**MMLU-math** Massive Multitask Language Understanding (MMLU) (Hendrycks et al., 2021a) is a benchmark designed to measure knowledge acquired by the LLMs. It covers 57 subjects. For MMLU-math, we choose the *abstract algebra, college mathematics, elementary mathematics, high school mathematics* subjects. Models need to think step by step and generate the final answer.

**MMLU-stem** We retain the original set of STEM in MMLU. Specifically, it include *abstract algebra, anatomy, astronomy, college biology, college chemistry, college computer science, college mathematics, college physics, computer security, conceptual physics, electrical engineering, elementary mathematics, high school biology, high school chemistry, high school computer science, high school mathematics, high school physics, high school statistics, machine learning*. Compare to MMLU-math, we use the probabilities of options to determine the answer instead of generate it, the options with highest probabilities among all the options will be considered as the final answer.

**NQ** NQ (Lee et al., 2019) is a benchmark for open-domain question answering derived from Google's Natural Questions dataset. The task is to predict a concise English answer to a question using only the information from English Wikipedia.

**SWAG** SWAG (Zellers et al., 2018) is a large-scale dataset for the task of grounded commonsense inference, unifying natural language inference and physically grounded reasoning. Each question is a video caption, with four answer choices about what might happen next in the scene. The correct answer is the (real) video caption for the next event in the video.

**MR-BEN-math** MR-BEN (Zeng et al., 2024) is a comprehensive benchmark demands a meta reasoning skill, where LMs are asked to locate and analyse potential errors in automatically generated reasoning steps. We choose the math among all subjects for evaluation.

---

**MR-BEN-math**

Following is a question and solution pair in subject college math. Your task is to examine the solutions step by step and determine the solution correctness. If the solution is incorrect, please further find out the first error step and explain the error reason.

<few-shot examples>

Below is the question and solution for you to solve:
Question: <question>
Options: <options>
Please follow the desired response format:
Solution Analysis: [Give a step by step analysis on the solution correctness here] Solution Correctness: [Input 'correct'/'incorrect' here to indicate the overall correctness of the solution]
First Error Step: [Input 'Step x' here to indicate the first error step here. Input 'N/A' if the solution is correct.]
Error Reason: [Input the error reason and the rectified reasoning of the first error step here. Input 'N/A' if the solution is correct.]

Please follow this format without any additional introductory or concluding statements.

---

**DocMath** DocMath (Zhao et al., 2024) is a benchmark specifically designed to evaluate the numerical reasoning capabilities of LLMs in the context of understanding and analyzing specialized documents containing both text and tables. Models are asked to generate answer through COT.

---

**DocMath**

You are a financial expert, you are supposed to answer the given question based on the provided financial document context. You need to first think through the problem step by step, documenting each necessary step. Then you are required to conclude your response with the final answer in your last sentence as 'Therefore, the answer is final answer'. The final answer should be a numeric value.

USER: <context and document>
Question: <question >
Let's think step by step to answer the given question.
ASSISTANT:

---

**ZebraLogic** ZebraLogic (Bill Yuchen Lin, 2024) is a benchmark consisting of Logic Grid Puzzles, assesses LLMs' logical reasoning capabilities. Each puzzle presents N houses with M features, requiring unique value assignments based on given clues. We use the average result of LLMs of different levels of puzzles.

---

**ZebraLogic**

A chat between a curious user and an artificial intelligence assistant. The assistant gives helpful, detailed, and polite answers to the user's questions.

USER:
# Puzzle to Solve
<puzzle>
## Clues:
<clues>
# Instruction
Now please solve the above puzzle. Present your reasoning and solution in the following json format:
<output format>

---

**LogiQA** LogiQA (Liu et al., 2020) is a benchmark which is sourced from expert-written questions for testing human Logical reasoning, covering multiple types of deductive reasoning.

**ProofWriter** Proofwriter (Tafjord et al., 2020) contains many small rulebases of facts and rules, expressed in English. Each rulebase also has a set of questions which can either be proven true or false using proofs of various depths, or the answer is "Unknown" or assumed negative.

ProofWriter

Task Description: You are given a problem description and a question. The task is to:
1) define all the predicates in the problem
2) parse the problem into logic rules based on the defined predicates
3) write all the facts mentioned in the problem
4) parse the question into the logic form
——
<few-shot examples>
——

Problem:
[[PROBLEM]]
Question:
[[QUESTION]
###

**GPQA**    GPQA (Rein et al., 2023) is a multiple-choice, Q&A dataset of very hard questions written and validated by experts in biology, physics, and chemistry.

**WinoGrande**    WinoGrande (Sakaguchi et al., 2021) is designed for commonsense reasoning. The samples are formulated as fill-in-the-blank questions where two answer choices are provided. The goal is to select the correct option based on commonsense knowledge.

**ARC-challenge**    AI2 Reasoning Challenge (ARC) (Clark et al., 2018) is a widely used dataset for evaluating large language models (LLMs) on their commonsense reasoning abilities. We choose the challenge set of ARC, which contains questions that simple retrieval or co-occurrence-based models struggle with, thus pushing models to reason more deeply.

**BBH**    BBH (Suzgun et al., 2022) is designed to evaluate LLMs' capability on difficult reasoning benchmarks, with a focus on symbolic reasoning. It includes problems that require manipulation of abstract symbols, helps to measure the generalization ability to symbolic reasoning beyond typical language tasks.

**MiniWob++**    MiniWob++ (Liu et al., 2018) is a browsers-based interactive tasks, include a range of tasks where an agent interacts with a simplified browser interface. Models are asked to generate the code for a goal-directed task in a simulation environment.

MiniWob++

You are required to navigate the web. To accomplish the task, use methods in Agent class to generate actions, with the following functions. type(characters: str): Type a string via the keyboard. click_xpath(xpath: str): Click an HTML element with a valid XPath. press(key_type: str): Press a key on the keyboard (enter, space, arrowleft, arrowright, backspace, arrowup, arrowdown, command+a, command+c, command+v). click_option(xpath: str): Click an option HTML element in a list with a valid XPath. movemouse(xpath: str): Move the mouse cursor on an HTML element with a valid XPath.

USER: The observation is:
<HTML description>
ASSISTANT: The action is:

