# OpenReview forum: "Math for AI: On the Generalization of Learning Mathematical Problem Solving"
_NeurIPS.cc/2024/Workshop/MATH-AI — MATH-AI 24_

### Official Review · Reviewer_VC9m · 2024-09-27
**Promising Insights with Room for Growth: Evaluating Mathematical Problem-Solving for AI Reasoning**

**Rating:** 8
**Confidence:** 3

**Review:**

This paper explores the use of mathematical problem-solving (MPS) data to enhance reasoning abilities in large language models (LLMs). The authors evaluated three training strategies: continual pretraining on mathematical text, instruction pretraining on synthetic QA pairs, and instruction tuning on benchmark datasets. The findings show that only continual pretraining on mathematical text reliably improves broader reasoning skills, while other methods fall short. Future work will focus on identifying data sources that can more effectively develop general reasoning abilities in AI models.

# Pros
## 1. Strong Writing:
The authors pose a clear research question and provide a thorough analysis of whether mathematical problem-solving enhances general reasoning abilities beyond just improving MPS benchmarks.

## 2. Valuable Findings:
They find that most open-source math datasets fail to support broader reasoning tasks, suggesting the need to shift research focus from “AI for math” to enhancing general reasoning skills.

## 3. Solid Experimentation:
The study effectively addresses overfitting by incorporating general instruction-following data into math-related training and adopting hybrid training.

# Cons
## 1. Lack of Solutions:
The paper identifies challenges but does not propose concrete solutions to address the limitations found.

---

### Official Review · Reviewer_SLug · 2024-10-07
**The paper examines whether mathematical problem-solving training for large language models enhances general reasoning skills, using three training strategies and rigorous methodologies, but could improve with more discussion on the mathematical concepts and dataset specifics**

**Rating:** 7
**Confidence:** 4

**Review:**

The authors investigates whether training large language models (LLMs) in mathematical problem-solving (MPS) can enhance general reasoning abilities beyond MPS benchmarks. The authors differentiate between “AI for math” and “math for AI,” highlighting the broader reasoning potential of the latter. They employ three training strategies—continual pretraining on mathematical texts, instruction pretraining on synthesized QA pairs, and instruction tuning on benchmark datasets—evaluating them across seven reasoning domains. The methodologies are rigorous and detailed. However, the paper would benefit from an overview discussion of the underlying mathematical concepts, more detailed information about the datasets used, and increased literature review for better contextualization.

---

### Official Review · Reviewer_VhvD · 2024-10-07
**Recommendation to accept**

**Rating:** 7
**Confidence:** 4

**Review:**

The work makes a concrete effort by determining when training LLM models on mathematical problem-solving (MPS) tasks extends their reasoning skills beyond strictly defined domains. The sufficiency of the paper in the testing of various types of reasoning algebra in logical, commonsense, and STEM reasoning indeed uses experimental evidence to enhance one’s comprehension of the effect of variances in mathematical training on general reasoning. This paper contributes to the understanding of the "math for AI" hypothesis since it shows that reasoning objectives are readily achieved after the continual pretraining of LLMS on mathematical texts. It assists in elucidating the use of mathematical problem solving data in training large language models and provides a definite path towards the enhancement of reasoning capacities of LLMs thereby making it an excellent candidate for acceptance.

---

### Decision · Program_Chairs · 2024-10-07

Accept